# Toxicity Assessment of Four Formulated Pyrethroid-Containing Binary Insecticides in Two Resistant Adult Tarnished Plant Bug (*Lygus lineolaris*) Populations

**DOI:** 10.3390/insects14090761

**Published:** 2023-09-13

**Authors:** Yuzhe Du, Yucheng Zhu, Shane Scheibener, Maribel Portilla

**Affiliations:** 1Southern Insect Management Research Unit, Agriculture Research Service, United States Department of Agriculture, 141 Experiment Station Road, Stoneville, MS 38776, USA; shane.scheibener@usda.gov (S.S.); maribel.portilla@usda.gov (M.P.); 2Pollinator Health in Southern Crop Research Service, Agriculture Research Service, United States Department of Agriculture, 141 Experiment Station Road, Stoneville, MS 38776, USA; yc.zhu@usda.gov

**Keywords:** *Lygus lineolaris*, pyrethroids’ binary mixtures, bioassay, resistance management, synergism, toxicity

## Abstract

**Simple Summary:**

This study evaluated the toxicity and resistance risks of four formulated pyrethroid-containing binary mixtures (Endigo, Leverage, Athena, and Hero) on susceptible and two resistant tarnished plant bug (TPB) populations, using a modified Potter Spray Tower. Among the binary mixtures, TPBs displayed the highest level of resistance to Hero and the lowest resistance towards Athena. A comparison of the binary mixture to their corresponding individual pyrethroid demonstrated significantly higher resistance ratios from Hero and Leverage, while Endigo and Athena displayed similar or lower resistance ratios in both resistant TPB populations. This study also assessed the interaction between the individual components in the binary mixtures using the calculated additive index (AI) and the co-toxicity coefficient (CTC). The two individual components in Endigo, Hero, and Athena exhibited synergistic interaction, whereas the components in Leverage (β-cyfluthrin and imidacloprid) exhibited an antagonistic interaction with both resistant TPB populations. Considering that Hero is a mixture of two pyrethroids, to which TPB may develop resistance easily, Endigo and Athena are likely superior products for slowing resistance development in TPB populations.

**Abstract:**

Over the past several decades, the extensive use of pyrethroids has led to the development of resistance in many insect populations, including the economically damaging pest tarnished plant bug (TPB), *Lygus lineolaris,* on cotton. To manage TPB resistance, several commercially formulated pyrethroid-containing binary mixtures, in combination with neonicotinoids or avermectin are recommended for TPB control and resistance management in the mid-South USA. This study aimed to evaluate the toxicity and resistance risks of four formulated pyrethroid-containing binary mixtures (Endigo, Leverage, Athena, and Hero) on one susceptible and two resistant TPB populations, which were field-collected in July (Field-R1) and October (Field-R2), respectively. Based on LC_50_ values, both resistant TPB populations displayed variable tolerance to the four binary mixtures, with Hero showing the highest resistance and Athena the lowest. Notably, the Field-R2 exhibited 1.5–3-fold higher resistance compared to the Field-R1 for all four binary insecticides. Moreover, both resistant TPB populations demonstrated significantly higher resistance ratios towards Hero and Leverage compared to their corresponding individual pyrethroid, while Endigo and Athena showed similar or lower resistance. This study also utilized the calculated additive index (AI) and co-toxicity coefficient (CTC) analysis, which revealed that the two individual components in Leverage exhibited antagonist effects against the two resistant TPB populations. In contrast, the two individual components in Endigo, Hero, and Athena displayed synergistic interactions. Considering that Hero is a mixture of two pyrethroids that can enhance the development of TPB resistance, our findings suggest that Endigo and Athena are likely superior products for slowing down resistance development in TPB populations. This study provides valuable insight for selecting the most effective mixtures to achieve better TPB control through synergistic toxicity analysis, while simultaneously reducing economic and environmental risks associated with resistance development in the insect pest.

## 1. Introduction

The tarnished plant bug (TPB) *Lygus lineolaris* (Palisot de Beauvois) (Hemiptera: Miridae) is a significant major pest in cotton-growing regions across the mid-Southern USA. With the eradication of the boll weevil and the adoption of Bt cotton to control lepidopteran pests, TPB infestations have increased yearly, resulting in severe damage to cotton fruiting buds [1]. Both TPB nymphs and adults use specialized piercing–sucking stylets and digestive enzymes to feed on cotton fruiting buds (squares) and small fruits (bolls), causing fruit abscission and damage to seeds and lint [2,3,4]. The prevalence of TPB adults is the highest during the pre-flowering stage of cotton, while nymphs are more commonly found during the flowering period. As a result, TPBs primarily inflict the most significant damage from the first square stage to the early flowering stages of cotton growth [2].

The use of insecticide has been the primary method for preventing and controlling TPB in the mid-south cotton growing area, with a variety of classes employed such as organophosphates, carbamates, neonicotinoids, pyrethroids, insect growth regulators, and sulfoximine [5,6,7,8,9]. Pyrethroids, which account for 30% of the global pesticide market, are synthetic insecticides based on natural pyrethrins found in *Chrysanthemum* flowers [10]. Over the past few decades, pyrethroids were extensively used to control agricultural crop pests and human disease vectors [11]. However, the overuse of pyrethroids led to the development of resistance in many insect pest populations, including TPBs [12]. Pyrethroid resistance in TPBs collected from cotton was first reported in Mississippi in 1993 [13,14], likely the result of selective pressure from early season insecticide applications targeting lepidopteran pests when TPBs were present. By 1999, resistance to pyrethroids in TPB was widespread in the mid-south region [15,16]. Pyrethroid resistance significantly reduced the effectiveness of chemical control and increased the cost and quantity of insecticides required to control this pest. Due to resistance development, pyrethroids are no longer recommended for TPB control in cotton in Mississippi [17].

To combat resistance, insecticide mixtures and rotation are proposed as important tools for resistance management. Currently, several commercially formulated pyrethroid binary mixtures in combination with neonicotinoids or avermectin are listed for TPB control and resistance management in Mississippi Delta region that include the following: Brigadier (bifenthrin + imidacloprid), Leverage (imidacloprid + β-cyfluthrin), Endigo (thiamethoxam + λ-cyhalothrin), Athena (bifenthrin + avermectin), and the two pyrethroid mixture Hero (bifenthrin + ζ-cypermethrin) (https://extension.msstate.edu/sites/default/files/publications/publications/P2471_web.pdf, accessed on 28 July 2022). Mixing insecticides with different modes of action is more effective in resistance management programs compared to rotational strategies [18]. Previously, mixtures consisted of a pyrethroid with carbamate [19] or a pyrethroid with organophosphorus [20,21,22,23]. More recently, the development of neonicotinoid insecticides with reduced toxicity to human compared to previously used organophosphates and carbamates can now provide broad spectrum control of numerous crop-damaging insects. Pyrethroids target insect voltage-gated sodium channels [11], and neonicotinoids act as agonists for the nicotinic acetylcholine receptors (nAChR), and both classes of insecticide impact the central nervous system of insects (Table 1) [24]. Mixtures of neonicotinoids and pyrethroids are useful for resistance management as highly effective tools against some of the world’s most destructive crop pests [25]. Avermectin, on the other hand, allosterically activates glutamate-gated chloride channels (GluCls) in insect nerve and muscle cells, causing cell hyperpolarization, eventually resulting in insect paralysis and death [26]. Pyrethroids can also be mixed with avermectin and applied for insect pest control. Insecticide mixtures afford two key advantages: targeting a broad spectrum of pest species and managing pesticide resistance.

Insecticide resistance research in the past predominantly focused on individual insecticides. Testing on commercialized formulated mixtures is not well studied, especially in resistant TPB populations. Here, we conducted a comprehensive investigation using two resistant field TPB populations collected in July and October, respectively, from wild-host plants in Coahoma County, a cotton growing area in the Northern Mississippi Delta region, USA. We conducted dose–response bioassays of four formulated pyrethroid-containing binary mixtures using a modified Potter Spray Tower. Additionally, we analyzed the potential interaction between the two individual components in the four binary mixtures on susceptible and two resistant TPB populations. Our findings provide valuable information for selecting the most effective mixtures to achieve better TPB control. By understanding the interactions between the individual components in the binary mixtures, we can optimize their use and develop targeted resistance management strategies.

## 2. Materials and Methods

### 2.1. Insect Populations

The laboratory susceptible TPB population (Lab-S) was collected from the hills and wooded area of Crossett, AR, where insecticides were infrequently used. This population is historically recognized as a susceptible one and was used in previous studies [8,15,27]. Lab-S TPB was reared on an artificial diet and without any exposure to insecticides, following the method outlined in Portilla et al. [28,29]. The resistant TPB population was collected from Coahoma County in July (Field-R1) and October 2022 (Field-R2), respectively, located in the northern Delta cotton growing region of Mississippi. For the bioassays, the TPB populations were maintained under laboratory conditions (27 ± 2 °C, 65 ± 10% RH and a 12:12 h (L:D) photoperiod) prior to and during the experiments.

### 2.2. Insecticides

The insecticides used in this study were all formulated and included the following: Endigo 2.06 ZC (Syngenta, Greensboro, NC, USA), Warrior II (Syngenta), Centric 40 WG (Syngenta), Leverage 360 EC (Bayer Crop Science, Durham, NC, USA), Baythroid XL (Bayer), Advise^®^ Four (Winfield, Philadelphia, PA, USA), Hero 1.24 (FMC, Philadelphia, PA, USA), Mustang^®^ Maxx (FMC), Tundra^®^ EC (Winfield), Athena (FMC), and Epi-Mek (Agri-Mek 0.15 EC, Syngenta). Insecticides were purchased from local agricultural suppliers near Leland, MS, with the manufacturers and mode of action provided in Table 1.

### 2.3. Laboratory Spray Tower Bioassays

Twenty TPB adults from the Lab-S (7–9 d old) or Field-R mixed age populations were placed into each 500 mL polypropylene cage (D × H: 9.3 × 10 cm). To provide air exchange, two 5 cm holes were made at the bottom and on the top of the cage and covered with fabric mesh. A modified spray tower with the original spray nozzle of a Potter Spray Tower (Burkard Scientific Ltd., Uxbridge, UK) was used in this study with settings of air pressure at 69 kPa or 10 psi, spray distance at 22 cm, and spray volume at 0.5 mL. Pesticide solutions were dissolved in deionized H_2_O and diluted serially to obtain the five desired concentrations. Two 7–8 cm long whole green beans, as food for caged bugs, were placed at the bottom of each cage. Pesticide solution was sprayed into the cage to cover the inner wall, green beans, and TPBs. Prior to spraying insecticide solutions, a control was conducted by spraying 0.5 mL of deionized water. After treatment, the caged TPB adults were maintained in an incubator set at 27 ± 2 °C, 65 ± 10% RH, and a 12:12 (L:D) photoperiod. Mortality was determined 48 h after the spray treatment. The treated adults were considered dead if the bugs were unable to walk or fly. Three or four replications (60–80 adults per concentration), depending on the availability of the bugs, were included for each treatment. Five concentrations (5, 10, 25, 50, 100 μg/mL) of Endigo, Leverage, Hero, and Athena were applied on the Lab-S TPB population, and higher concentrations of 50, 100, 200, 400, and 800 μg/mL or 100, 200, 400, 800, and 1600 μg/mL were used for Field-R1 or Field-R1 TPB population, respectively.

### 2.4. Evaluation of Potential Interaction of Two Individual Component in Formulated Binary Mixtures

To analyze the potential interaction of two individual components in four binary insecticides, the synergistic, additive, or antagonistic effects in four formulated binary mixtures (Endigo, Leverage, Hero, and Athena) were assessed using Marking’s additive index (AI) methods [30] and Sun and Johnson [31] methods, which are based on the LC_50_ values from individual pesticides and their mixtures to characterize the interactions of pesticides. The AI was adopted to assess binary toxicity as follows:$$S=\left(\frac{Am{LC}_{50}}{Ai{LC}_{50}}\right)+\left(\frac{Bm{LC}_{50}}{Bi{LC}_{50}}\right)$$
where *S* represents the sum of the toxicity of pesticides *A* and *B*; *A*m indicates the LC_50_ of pesticide *A* in mixture; *A*i is the LC_50_ of individual pesticide *A*; *B*m indicates the LC_50_ of pesticide *B* in mixture; and *B*i is the LC_50_ of individual pesticide *B*.

The AI value was determined from the sum of *S* based on the appropriate formulas as follows:$$\mathrm{AI}=\left(\frac{1}{S}\right)-1\;\mathrm{for}\;S<1.0\;\mathrm{and}\;\mathrm{AI}=1-S\;\mathrm{for}\;S\geq 1.0$$

Binary toxicities were classified as antagonistic effect (AI ≤ −0.2), additive (−0.2 < AI ≤ 0.25), or synergistic effect (AI > 0.25) accordingly. The greater the AI value, the greater the pesticide synergy [32].

Sun and Johnson methods [31]:(1)Toxicity Index (T.I.) (using A as standard, A and B are individual components in formulated mixture) T.I. of A = 100 and T.I. of B = ALC_50_/BLC_50_ × 100.(2)Actual Toxicity Index (ATI) of mixture (using A as standard) ATI = LC_50_ of A/LC_50_ of (A + B) × 100.(3)Theoretical Toxicity Index (TTI) of mixture: TTI = T.I. of A × % of A in mixture + T.I. of B × % of B in mixture. From the actual and theoretical toxicity of the mixture, the mixture toxicity can be calculated by the following CTC equation.(4)CTC (co-toxicity coefficient): CTC = ATI/TTI × 100. When the co-toxicity coefficient of the formulated mixture is 100, the effect of this mixture indicates the probability of similar action. If the mixture results in a coefficient significantly greater than 100, it indicates a synergistic action.

### 2.5. Data Analysis

Bioassay data were analyzed using Probit analysis using SPSS software (version 19.0, SPSS Inc., Chicago, IL, USA, 2003). LC_50_ values of different populations were considered significantly different if 95% confidence intervals were not overlapping. Resistance ratios (RR) were calculated as the ratio of LC_50_ value of the resistant populations (Field-R1/Field-R2) to that of the Lab-S TPB populations. The comparison of resistance ratio (RR) was conducted by plotting using JMP software with one-way analysis of variance followed by Tukey’s HSD to determine statistical significance (*p* < 0.05).

## 3. Results

### 3.1. Toxicity of the Four Formulated Mixtures against the Laboratory Susceptible Strain

Four pyrethroid-containing formulated binary insecticides (Endigo, Leverage, Athena, and Hero) were applied to Lab-S and two field-collected resistant TPB populations (Field-R1 and Field-R2). Following exposure to spray treatment for 48 h, the LC_50_ values for Endigo, Leverage, Hero, and Athena in Lab-S TPB were 22.54, 20.53, 20.39, and 73.60 µg/mL, respectively (Table 2).

### 3.2. Toxicity of Formulated Mixtures against the Two Field-Collected Resistant TPB Populations

In contrast, the Field-R1 TPB population collected from Coahoma County, MS, in July 2022 displayed significantly higher resistance to the four binary insecticides. The LC_50_ values for this population were significantly elevated to 166.64, 273.44, 329.96, and 479.60 µg/mL for Endigo, Leverage, Hero, and Athena, respectively (Table 2). The corresponding resistance ratios (RR) for each binary mixture were 7.39-fold for Endigo, 13.32-fold for Leverage, 16.18-fold for Hero, and 6.56-fold for Athena (Table 2, Figure 1a). Among the four binary insecticides, Hero exhibited the highest RR, while Athena showed the lowest RR (Table 2, Figure 1a).

Additionally, another Field-R2 TPB population collected from Coahoma County, MS, in October 2022 also demonstrated increased resistance to the four binary insecticides. The LC_50_ values for this population were significantly elevated to 315.18, 513.94, 970.67, and 708.30 µg/mL for Endigo, Leverage, Hero, and Athena, respectively (Table 2). The corresponding resistance ratios (RR) for each binary mixture were 13.98-fold for Endigo, 25.03-fold for Leverage, 47.60-fold for Hero, and 9.62-fold for Athena (Table 2, Figure 1b). Similar to the Field-R1 TPB population, Hero exhibited the highest RR, while Athena displayed the lowest RR in Field-R2 TPB as well (Table 2, Figure 1b). Furthermore, when comparing the Field-R2 TPB population collected in October with the Field-R1 collected in July, Endigo and Leverage showed approximately 2-fold higher resistance, Hero’s resistance increased up to 3-fold, whereas Athena displayed only 1.4-fold increase in resistance (Table 2, Figure 1).

### 3.3. Toxicity Comparison of the Individual Pyrethroid with the Binary Mixture

The bioassay of the corresponding individual components and binary mixture insecticides were examined simultaneously. For the Field-R1 TPB resistant population, both Leverage and Hero exhibited significant higher resistance levels compared to the corresponding individual pyrethroids present in their formulations (Figure 1a). In contrast, Endigo and Athena displayed similar resistance levels to the corresponding individual pyrethroid (Figure 1a).

Similarly, for the Field-R2 TPB population, Endigo showed a similar resistance level to the individual λ-cyhalothrin (Figure 1b). However, Leverage and Hero displayed significantly higher resistance levels compared to their respective individual pyrethroids. On the other hand, Athena exhibited a significant lower resistance level to the individual pyrethroid bifenthrin in its formulation (Figure 1b).

### 3.4. Analysis of the Potential Interaction of Two Individual Components in the Four Formulated Binary Mixtures

To analyze the potential interaction of the two individual components in the four formulated binary mixtures, we employed the additive index (AI) and co-toxicity coefficient (CTC) analysis. According to calculated AI value (AI > 0) and CTC value (CTC > 100), synergistic effects were determined on the Lab-S strain for all four formulated pyrethroid-containing binary insecticides (Table 3).

Calculated AI (AI > 0) and CTC values (CTC > 100) for Endigo and Athena also indicated synergistic interaction between the two components in both Field-R1 and Field-R2 TPB populations (Table 3). Although Hero showed the highest resistance ratio among all four tested pesticides, two pyrethroids in it exhibited a synergistic effect with an AI value of 1.14 for Field-R1 and 0.55 for Field-R2, or CTC value of 213.6 for Field-R1 and 154.9 for Field-R2, respectively (Table 3). However, calculated AI value (−0.38 for Field-R1 and −0.28 for Field-R2), or CTC value (72.6 for Field-R1 and 88.0 for Field-R2) for Leverage indicated antagonistic interaction between imidacloprid and β-cyfluthrin in Leverage for both Field-R1 and Field-R2 TPB populations (Table 3).

## 4. Discussion

The primary objective of this study was to evaluate the toxicity and resistance levels in two field-resistant TPB populations against four formulated pyrethroid-containing binary insecticides through spray treatments. Our results elucidated significant resistance development in both Field-R1 and Field-R2 TPB populations to the four binary insecticides. Among the tested four binary formulations, TPBs consistently exhibited the highest resistance ratio towards Hero, indicating greater tolerance to this mixture in both Field-R1 and Field-R2 TPB populations. On the other hand, TPBs displayed the lowest resistance ratio to Athena and highlights its continued effectiveness against resistant TPB populations. Interestingly, the Field-R2 TPB population collected in October displayed about 1.4–3-fold increase in resistance compared to the Field-R1 TPB population collected in July. These findings align with our previous investigations on individual insecticides examined in both resistant TPB populations, which suggested that the resistance levels in TPB populations escalated and resulted in a rapid evolution of insecticide resistance due to selection pressure accumulated within a single growing season [33].

Among the four pyrethroid-containing binary insecticides tested, both Field-R1 and Field-R2 TPB populations exhibited higher resistance levels towards Leverage and Hero compared to their corresponding individual pyrethroid. On the other hand, both field-resistant TPBs exhibited similar or lower resistance to Endigo and Athena. These findings suggested that Endigo and Athena are likely superior products for slowing down resistance development compared to their corresponding individual pyrethroid in resistant TPB populations.

In the current study, the utilization of the calculated additive index (AI) and co-toxicity coefficient (CTC) assessed the interaction between the individual components in the four binary mixtures. Results indicated that the two individual components in Endigo, Leverage, Hero, and Athena exhibited synergistic effects in the Lab-S TPB population, and increased pyrethroid toxicity when mixed with neonicotinoids or avermectin. Previous studies also reported synergistic interaction between pyrethroids and neonicotinoid insecticides in binary mixtures for other insect pest species, such as mosquitos [34,35,36], *Cimex lectularius* [37], *Cimex hemipterus* [25], *Bombyx mori* [38], and *Drosophila* [39]. Corbett [19] proposed a general theory to explain the synergistic interactions among different insecticide mixtures. According to this theory, one insecticide in the mixture interferes with the metabolic detoxification of the other insecticide, thereby enhancing the toxicity of the latter one. Applications of either Endigo or Leverage on TPBs bind both pyrethroids and neonicotinoids to the monooxygenase P450 [40,41], which hydrolyzes cytochrome P450 to catalyze both insecticides. Consequently, this prevents subsequent binding of neonicotinoid or pyrethroid insecticides catalyzed by monooxygenase enzymes [42]. In the case of Athena, the mixture of the pyrethroid bifenthrin and abamectin may allosterically activate the glutamate-gated chloride channel (GluCl) combined with bifenthrin, leading to synergistic action [26].

In addition, the calculated additive index (AI) and co-toxicity coefficient (CTC) also indicated two individual components in Endigo, Hero, and Athena (but not Leverage) exhibited synergistic effects against both Field-R1 and Field-R2 TPB populations. Furthermore, both resistant TPB populations showed similar or lower resistance levels for Athena and Endigo than their corresponding individual pyrethroid. Pre-mixed binary insecticides that contain two different active ingredients (i.e., different modes of action and target sites) may be more likely to induce synergistic toxicity, facilitate uptake, and slow the development of resistance [39,42]. Endigo and Athena contain two components with differing modes of action and target either the nicotinic acetylcholine receptor (nAChR) or glutamate-gated chloride channel (GluCl), respectively (Table 1). The probability of cross-resistance between pyrethroids and neonicotinoids or avermectin is low, and the development of multiple resistance based on different resistance alleles is also low; thus, individuals with multiple resistance mechanisms will be rare. This may explain the observed similar or lower resistance levels in resistant TPBs exposed to Endigo and Athena compared to their corresponding individual pyrethroid in their formulation. However, resistant TPBs exposed to Hero, a mixture of bifenthrin and ζ-cypermethrin with the same mode of action (targeting sodium channels, Table 1), exhibited much higher resistance ratio than the corresponding individual components. Developed resistance to multiple insecticides under long-term selection pressure in the field may explain the field-collected TPBs with higher resistance to Hero. An increase in insecticide applications rates in cotton producing areas of the Delta since 1999 (0.3 applications/year) may also contribute to the observed resistance [43]. Even though Hero induced synergism in both resistant TPB populations, the probability of cross-resistance between bifenthrin and ζ-cypermethrin is high [17], and led to the highest observed resistance ratio (47.60-fold) among all tested insecticides. Therefore, Hero is not a suitable choice for long-term TPB multiple resistance management, while Endigo and Athena are more effective and recommended for such management. Also, TPBs displayed a higher resistance level towards Leverage than the corresponding individual pyrethroid β-cyfluthrin and demonstrated antagonistic action in the two resistant TPB populations. The more recent intensive use of the individual component in Leverage, imidacloprid, to control sucking insect pests may explain the increased resistance to the binary mixture [44]. Although Brigadier (bifenthrin + imidacloprid) was not tested on TPBs in this study, previous research showed that resistant TPBs exposed to tank mixtures of bifenthrin and imidacloprid exhibited antagonistic effects consistent with the antagonism observed with Leverage currently [44]. Moreover, multiple resistance mechanisms, including decreased cuticular penetration and target site insensitivity, may contribute to the antagonistic effect as well [42].

Although manufacturers may optimize component ratios in pyrethroid-containing binary insecticides (e.g., Endigo, Leverage, and Athena), tank mixing of insecticides is common practice in integrated pest management for row crops where multiple insects and plant pathogens are targeted in the field [45]. For instance, tank mixing bifenthrin plus acephate is synergistic on TPBs and is often used to control TPBs and other cotton pests [44]. Additionally, antagonistic interactions between tank mixing pyrethroids and organophosphates have been reported in other insects, such as *Bemisia tabaci* [46]. Similarly, a binary mixture of deltamethrin and chlorpyrifos showed antagonistic effects against bollworm at 96 h [22]. While tank mixing lacks an optimized ratio, the interaction of bifenthrin and fipronil was additive on *Musca domestica* when in the ratio 1:1, whereas when used in the ratio of LC_50_:LC_50_, the mixtures produced a synergistic effect [42]. Therefore, the antagonistic or synergistic interactions among tank mixed binary insecticides depends on the type of insecticides used, the ratios of components, and genetic background of the test organism [42].

The use of binary mixtures is a common practice for controlling resistant insect species, and two of the mixtures in the current study are recommended options for managing resistant TPB populations. Our study highlights the importance of understanding the interactions between different insecticides to develop effective pest management strategies. Considering the control cost, two or more compounds might be mixed to obtain the best control effects [35]. However, long-term use of insecticide mixtures can pose risk to beneficial organisms [47]. Some mixtures including neonicotinoids are toxic to the bee *Apis mellifera* [48,49], while others can disrupt the survival and development of non-target aquatic organisms [50,51,52]. Despite these risks, our study found that two formulated pyrethroid-containing mixtures Endigo and Athena exhibited synergistic interactions between pyrethroid and non-pyrethroid mixtures, which could effectively manage resistant TPB populations. Our work contributes to the development of more efficient and sustainable approaches for combating TPB resistance and enhancing overall pest management strategies in cotton cultivation.

It is important to note that the interaction between the chemical components of pesticide mixtures can also be affected by the inclusion of adjuvants or specific formulas. Adjuvants are added to mixtures for specific purposes and can have an impact on how the chemical components interact with each other [53]. For instance, wetting agents are commonly added to some synthetic pyrethroids to improve their spreading and cuticle penetration, but they may not be an effective adjuvant for abamectin [53]. The synergistic or antagonistic effect observed between two pesticide components may also be influenced by the interaction between their adjuvants. However, in this study, our examinations mainly focused on the active ingredients of the four commercially formulated pyrethroid-containing binary insecticides. How adjuvants influence the efficacy of pesticides, especially in mixtures, will be our consideration in future studies to better understand all influencing factors on pesticide resistance development in the TPB.

## Figures and Tables

**Figure 1 insects-14-00761-f001:**
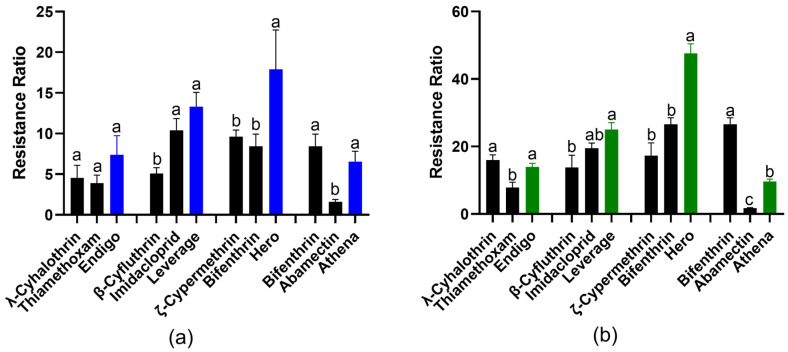
The resistance ratios (RR) of formulated pyrethroid-containing binary insecticides (colored bars) and corresponding individual components (black bars) in two field-resistant TPB populations. (**a**) Field-R1, collected in July, and (**b**) Field-R2, collected in October, after spray application for 48 h. Resistance ratios were determined as the ratio of LC_50_ of the field-resistant TPB population (Field-R1/Field R2) divided by LC_50_ of the Lab-S strain. LC_50_ values and 95% confidence intervals were calculated with Probit analyses using SPSS software and data were presented as the means ± S.D. Within each group of resistance ratio, means sharing different letter on the top of bars are significantly different, as determined using one-way analysis of variance with Tukey’s HSD test, and significant values were set at *p* < 0.05.

**Table 1 insects-14-00761-t001:** The commercial name, common name (with percentage active ingredient), manufacturer, and mode of action (MOA) for insecticides used in the current study.

	Commercial Name	Common Name (Percentage of Active Ingredient)	Manufacturer	Mode of Action
1	Endigo 2.06 ZC	Thiamethoxam (12.60%) + λ-Cyhalothrin (9.48%)	Syngenta	3A + 4A
2	Warrior II	λ-cyhalothrin (22.8%)	Syngenta	3A
3	Centric 40 WG	Thiamethoxam (40%)	Syngenta	4A
4	Leverage 360 EC	Imidacloprid (21.0%) + β-Cyfluthrin (10.5%)	Bayer Crop Science	4A + 3A
5	Baythroid XL	β-cyfluthrin (12.7%)	Bayer	3A
6	Advise^®^ Four	Imidacloprid (40.4%)	Winfield	4A
7	Hero 1.24	Bifenthrin (11.25%) + ζ-Cypermethrin (3.75%)	FMC	3A + 3A
8	Tundra^®^ EC	Bifenthrin (25.1%)	Winfield	3A
9	Mustang Maxx	ζ-cypermethrin (9.15%)	FMC	3A
10	Athena	Bifenthrin (8.84%) + Avermectin B1 (1.33%)	FMC	3A + 6
11	Epi-Mek	Abamectin (15%)	Syngenta	6

**Table 2 insects-14-00761-t002:** The toxicity of formulated pyrethroid-containing binary insecticides against the susceptible and two resistant populations of *Lygus lineolaris* after spray application for 48 h.

Compounds	Strain ^a^	Slope	LC_50_ (μg/mL) ^b^	95% Confidence Limits (μg/mL)	χ^2^	*p*	RR ^c^
Endigo	Lab-S	2.877 ± 0.296	22.54	19.19–26.49	1.53	0.68	―
	Field-R1	2.364 ± 0.562	166.64	113.75–242.05	4.74	0.32	7.39
	Field-R2	1.947 ± 0.334	315.18	254.95–397.47	1.07	0.90	13.98
Leverage	Lab-S	2.613 ± 0.278	20.53	17.35–24.16	3.27	0.35	―
	Field-R1	5.049 ± 0.911	273.44	237.29–318.85	2.39	0.30	13.32
	Field-R2	1.628 ± 0.346	513.94	384.53–883.0	1.75	0.78	25.03
Hero	Lab-S	2.929 ± 0.285	20.39	17.59–23.66	2.20	0.53	―
	Field-R1	2.308 ± 0.427	329.96	266.61–439.07	2.11	0.35	16.18
	Field-R2	3.377 ± 0.411	970.67	836.94–1177.82	5.72	0.33	47.60
Athena	Lab-S	2.876 ± 0.351	73.60	62.84–86.62	2.74	0.26	―
	Field-R1	3.363 ± 0.648	479.60	389.88–598.0	1.07	0.59	6.56
	Field-R2	3.283 ± 0.403	708.30	619.44–828.43	6.70	0.15	9.62

^a^ Lab-S: laboratory-reared susceptible strain; Field-R: field-collected resistant TPB populations, collected in July (Field-R1) and October (Field-R2). ^b^ LC_50_ values and 95% confidence intervals were calculated by Probit analyses using SPSS software. ^c^ Resistant ratio (RR) calculated by dividing LC_50_ of Lab-S by LC_50_ of field-collected resistant TPB populations.

**Table 3 insects-14-00761-t003:** The calculated additive index (AI) and co-toxicity coefficient (CTC) value of binary insecticides in the Lab-S and two field-resistant *Lygus lineolaris* populations (Field-R1 and Field-R2) after spray application for 48 h.

Compounds	Strain	AI	AI (Confidence Interval)	CTC	CTC (Confidence Interval)
Endigo	Lab-S	1.28	0.94–1.67	227.7	174.0–267.4
	Field-R1	0.26	−0.12–0.85	126.4	87.0–185.2
	Field-R2	0.79	0.42–1.21	157.4	124.8–194.6
Leverage	Lab-S	0.32	0.12–0.56	131.6	111.8–155.7
	Field-R1	−0.38	−0.60–−0.19	72.6	62.3–83.7
	Field-R2	−0.28	−0.58–−0.04	88.0	51.2–117.6
Hero	Lab-S	2.71	2.18–3.28	369.3	318.2–428.0
	Field-R1	1.14	0.60–1.64	213.6	160.6–264.4
	Field-R2	0.55	0.28–0.80	154.9	127.7–179.7
Athena	Lab-S	1.22	0.89–1.61	222.5	189.1–260.6
	Field-R1	1.41	0.87–1.88	233.8	187.5–287.5
	Field-R2	2.16	1.70–2.61	316.4	270.5–361.8

## Data Availability

The data that support the findings of this study are available on request from the corresponding author.

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
