# Peer review of "Toxicity Assessment of Four Formulated Pyrethroid-Containing Binary Insecticides in Two Resistant Adult Tarnished Plant Bug (Lygus lineolaris) Populations"

_insects, 2023, doi:10.3390/insects14090761_

Round 1

Reviewer 1 Report

This manuscript evaluated the toxicity and resistance risks of four formulated pyrethroid-containing binary mixtures (Endigo, Leverage, Athena, and Hero) on one susceptible and two resistant TPB populations, and assessed the interaction between the individual components in the binary mixtures. The overall structure of the article is relatively complete and the data is sufficient, which provides valuable insight for selecting the most effective mixtures to achieve better TPB control. However, there are still some problems that need to be addressed before publication.

1.     Suggest L75-“To combat resistance …” to L98-“…and managing pesticide resistance.” as another paragraph

2.     L136: “The TPB adults from the lab-S (7-9 d old) or field-R mixed age populations were placed into plastic cups…”- Why the age of the test insects in the laboratory population and the field population is different, whether the results will be affected

3.     L142: “spray distance”- A space needs to be added after this word

4.     L142-144: “Pesticide solutions were dissolved in deionized H2O and diluted serially to obtain the five desired concentrations.”- The concentration gradients of different pesticide treatments in the bioassay need to be listed

5.     L181: “(4.)” change to “(4).”

6.     L245-249: “For the Field-R1 TPB resistant population…Endigo and Athena displayed similar resistance levels to the corresponding individual pyrethroid (Figure 1A) [33]” Are these results obtained from this study experiment, and why should other literature be cited? Suggest discussing with other research findings in the discussion section not in result

7.     L251-253: “However, Leverage and Hero displayed significantly higher resistance levels compared to their respective individual pyrethroids (unpublished data).” From Figure 1B of this study, it can be concluded why it is necessary to indicate the use of unpublished data

8.     L288: There is an extra space after the word “populations”, please delete

9.     L181: “spodoptera littoralis” change to “Spodoptera littoralis

10.  References: Latin names of species should be italicized. For example, L431-“Culex quinquefasciatus”; L461-“Danio rerio”; L478-“Bombyx mori”; L487-“Musca domestica”; L489-“Lygus lineolaris”. Please check and unify the format of references according to the requirements of the journal

Author Response

This manuscript evaluated the toxicity and resistance risks of four formulated pyrethroid-containing binary mixtures (Endigo, Leverage, Athena, and Hero) on one susceptible and two resistant TPB populations, and assessed the interaction between the individual components in the binary mixtures. The overall structure of the article is relatively complete and the data is sufficient, which provides valuable insight for selecting the most effective mixtures to achieve better TPB control. However, there are still some problems that need to be addressed before publication.

  1. Suggest L75-“To combat resistance …” to L98-“…and managing pesticide ” as another paragraph

Revised as suggested.

  1. L136: “The TPB adults from the lab-S (7-9 d old) or field-R mixed age populations were placed into plastic cups…”- Why the age of the test insects in the laboratory population and the field population is different, whether the results will be affected

Response: Snodgrass (J. Econ. Entomol. 1996, 89(5): 1053-1059) tested age effect on permethrin susceptibility only on lab colony. He showed that 2-3 day old bugs had similar mortality as 9-10 day old bugs, but 16-17 day old bugs had higher mortality than 2-3 and 9-10 day old bugs. He suggested to use <10 day old bugs for lab strain. He also indicated “The age of field-collected plant bugs cannot be determined unless nymphs are collected and reared in the laboratory”. We usually judge bug’s vigor by visual observation of how quick the bugs can craw and fly out sweeping net and cage.

  1. L142: “spray distance”- A space needs to be added after this word

Revised as suggested. Thanks.

  1. L142-144: “Pesticide solutions were dissolved in deionized H2O and diluted serially to obtain the five desired

concentrations.”- The concentration gradients of different pesticide treatments in the bioassay need to be listed

The concentrations were added.

  1. L181: “(4.)” change to “(4).”

Revised as suggested. Thanks.

  1. L245-249: “For the Field-R1 TPB resistant population…Endigo and Athena displayed similar resistance levels to the corresponding individual pyrethroid (Figure 1A) [33]” Are these results obtained from this study experiment, and why should other literature be cited? Suggest discussing with other research findings in the discussion section not in result

Removed [33].

  1. L251-253: “However, Leverage and Hero displayed significantly higher resistance levels compared to their respective individual pyrethroids (unpublished data).” From Figure 1B of this study, it can be concluded why it is necessary to indicate the use of unpublished data

“(unpublished data)” was removed.

  1. L288: There is an extra space after the word “populations”, please delete

Revised. Thanks.

  1. L181: “spodoptera littoralis” change to “Spodoptera littoralis

Reformatted the name on L442. Thanks.

  1. References: Latin names of species should be italicized. For example, L431-“Culex quinquefasciatus”; L461- “Danio rerio”; L478-“Bombyx mori”; L487-“Musca domestica”; L489-“Lygus lineolaris”. Please check and unify the format of references according to the requirements of the journal

Revised. Thanks! Also revised Drosophila on L480.

We appreciate reviewers for their comments and suggestions.

Reviewer 2 Report

The overuse of pyrethroids led to the development of resistance in many insect pest populations. To combat resistance, insecticide mixtures, and rotation are proposed as essential tools for resistance management. Du et al., through bioassay, assessed the interaction between the individual components in the binary mixtures using the calculated additive index (AI) and the co-toxicity coefficient (CTC). The two separate components in Endigo, Hero, and Athena exhibited synergistic interaction. Endigo and Athena are likely superior products for slowing resistance development in TPB populations. I suggest the paper might be made acceptable pending revisions based on the comments below.

1.What is meant by--in Table 2? The compounds > Insecticides.

2. In the text, it is “Figure 1A, Figure 1B”, while in the figures it is “1a, 1b”. Please be consistent.

3. Line 304-305, note the use of semicolons and commas.

4. Please check the use of singular and plural in the text. Line255: change " component" to " components".

The grammar and sentence structure of this papar is commendable.

Author Response

The overuse of pyrethroids led to the development of resistance in many insect pest populations. To combat resistance, insecticide mixtures, and rotation are proposed as essential tools for resistance management. Du et al., through bioassay, assessed the interaction between the individual components in the binary mixtures using the calculated additive index (AI) and the co-toxicity coefficient (CTC). The two separate components in Endigo, Hero, and Athena exhibited synergistic interaction. Endigo and Athena are likely superior products for slowing resistance development in TPB populations. I suggest the paper might be made acceptable pending revisions based on the

comments below.

  1. What is meant by“--” in Table 2? The “compounds” > “Insecticides”.

“--” was replaced by “―” which is the baseline (LC50 of susceptible strain) for calculating resistance ratio (RR).

  1. In the text, it is “Figure 1A, Figure 1B”, while in the figures it is “1a, 1b”. Please be

Revised to 1a and 1b. Thanks.

  1. Line 304-305, note the use of semicolons and

Revised. Thanks.

  1. Please check the use of singular and plural in the Line255: change " component" to " components".

Revised. I used Editor (MS Office 365) and found another grammar error.

Reviewer 3 Report

AI and CTC are calculated as point estimators without confidence intervals, while LC50 is accompanied by confidence interval calculation. It would be helpful for the readers to add an approximation to confidence interval estimation. This might be accomplished by simulation, or by error propagation approach; you can find a guideline at https://www.statisticshowto.com/statistics-basics/error-propagation/.

Overall, I like your approach. Given the economic damages by the tarnished plant bug, your contribution will be of great practical significance to farmers. Maybe, not only in cotton.

Only restriction is, that at least a part of the effects may not only be related to the active compounds used. Surfactants might play a significant role, too. At least, this aspect should be mentioned in the course of your discussion.

Your discussion of tank mixture aspects is correct, but fails to mention the (unclear) role of surfactants and additives. 

Author Response

AI and CTC are calculated as point estimators without confidence intervals, while LC50 is accompanied by confidence interval calculation. It would be helpful for the readers to add an approximation to confidence interval estimation.

This might be accomplished by simulation, or by error propagation approach; you can find a guideline at https://www.statisticshowto.com/statistics-basics/error-propagation/.

 The confidence intervals of AI and CTC were added in Table 3

Overall, I like your approach. Given the economic damages by the tarnished plant bug, your contribution will be of great practical significance to farmers. Maybe, not only in cotton.

Thank you very much for your encouragement!

Only restriction is, that at least a part of the effects may not only be related to the active compounds used. Surfactants might play a significant role, too. At least, this aspect should be mentioned in the course of your discussion.

Response: We added a paragraph to the end of the Discussion section to discuss the potential interaction between insecticide and adjuvant.

Your discussion of tank mixture aspects is correct, but fails to mention the (unclear) role of surfactants and additives.

Response: We added a paragraph to the end of the Discussion section to discuss the potential interaction between insecticide and adjuvant. Thank you very much!